cognition, behaviour, neuroscience

numerosity perception, numerical cognition, math abilities, texture density, magnitude perception

**Author for correspondence:**
D. C. Burr
e-mail: davidcharles.burr@unifi.it

# Spontaneous perception of numerosity in pre-school children

G. Anobile[1], G. Guerrini[2], D. C. Burr[2,3], M. Monti[2], B. Del Lucchese[1] and G. M. Cicchini[3]

[1]Department of Developmental Neuroscience, IRCCS Stella Maris Foundation, Pisa, Italy
[2]Department of Neuroscience, Psychology, Pharmacology and Child Health, University of Florence, Florence, Italy
[3]Institute of Neuroscience, National Research Council, Pisa, Italy

GA, 0000-0003-2796-0661; DCB, 0000-0003-1541-8832; GMC, 0000-0002-3303-0420

There is strong evidence that humans can make rough estimates of the numerosity of a set of items, almost from birth. However, as numerosity covaries with many non-numerical variables, the idea of a direct number sense has been challenged. Here we applied two different psychophysical paradigms to demonstrate the spontaneous perception of numerosity in a cohort of young pre-school children. The results of both tasks showed that even at that early developmental stage, humans spontaneously base the perceptual choice on numerosity, rather than on area or density. Precision in one of these tasks predicted mathematical abilities. The results reinforce strongly the idea of a primary number sense and provide further evidence linking mathematical skills to the sensory precision of the spontaneous number sense, rather than to mechanisms involved in handling explicit numerosity judgements or extensive exposure to mathematical teaching.

## 1. Introduction

Understanding numerosity processing is important for many reasons, not least because it predicts mathematical abilities [1–9], which has clear clinical, educational and economical implications [10]. Signatures for a number sense have been found even in newborns [11], opening the suggestive hypothesis that it serves as an early toolkit for the acquisition of arithmetic skills later in life [3,12].

Measuring directly the number sense can be troublesome, as numerosity intrinsically covaries with several other variables, such as area and density, making it difficult to ascertain what drives numerical judgements in comparison and estimation tasks. At the same time, the correlation between formal mathematics and numerosity proficiency may not reflect a direct link as the two tasks necessarily share cognitive resources for language-based numerical reasoning. A recent influential developmental theory has further suggested that humans are not born with a specific 'number sense', but that this is moulded from a more generalized 'magnitude-sense' after experience [13]. Others suggest that even during adulthood, numerosity is derived indirectly from other features, such as texture density and area [14,15].

Cicchini *et al.* [16] measured the relative salience of numerosity, area and density with an implicit task where subjects were simply asked to select the odd stimulus out of three. The results showed that adult observers spontaneously base their decisions on numerosity rather than area or density. Similarly, when reproducing dot arrays, adults and adolescents are more precise in matching numerosity rather than the other dimensions [17]. These results suggest that, at least from adolescence on, human perception of numerosity is direct and not subordinate to area and density estimation.

Here we show that young children, including pre-schoolers, also spontaneously base perceptual choices on numerosity, rather than on density or

area. We also demonstrate a strong link between numerosity judgements and mathematical abilities.

## 2. Material and methods

### (a) Participants

A total of 101 children (age range: 5.0–7.75, mean 6.1, s.d. 0.57) participated, of whom 79 were pre-schoolers (age range: 5–6.75, mean 5.75, s.d. 0.34) and 22 first graders (age range: 6.33–7.75, mean 6.75, s.d. 0.4). Formal mathematical abilities were measured with an age-standardized battery on 75 pre-schoolers and 22 first graders. Fifty-eight pre-schoolers performed the explicit two-alternative-forced-choice (2AFC) number discrimination, and 50 pre-schoolers the dot-array reproduction task. Twenty pre-schoolers and 22 first graders performed the three-alternative-forced-choice (3AFC) odd-one-out task. Participants had no medical or psychiatric diagnosis, as reported by parents and teachers. Participants were tested individually in a quiet room at school. The study was approved by the regional ethics committee at the *Azienda Ospedaliero-Universitaria Meyer* (protocol code: GR-2013-02358262). Parents signed the appropriate informed consent.

### (b) Sample size

We ran two separate *a priori* power analyses for testing the angle of maximal sensitivity and the correlation between psychophysical sensitivity and mathematical scores. For the first analysis, we aimed to provide a measure of the angle of maximal sensitivity with a 90% confidence interval that did not exceed $10°$ (*i.e.* a standard error less than $6°$). Previous research reported a population of adolescents having a standard deviation of $9°$ [17]. We assumed that the expected standard deviation of our younger population would be three times higher ($27°$), which suggests that at least 20 subjects are needed to meet the criterion on confidence interval (CI). In the second analysis, we targeted a power of 0.8 for a significant correlation between perceptual sensitivity and mathematical abilities ($\alpha = 0.05$ one-tailed). According to a recent meta-analysis [2], the effect size is expected to be 0.4. This suggested a sample size of 32 participants. Analysis was performed by G*POWER software.

### (c) Stimuli

Stimuli were generated with the Psychophysics Toolbox for MATLAB and presented at a viewing distance of 57 cm on a 23 inch LCD monitor (1920 × 1080 pixels, 60 Hz). Stimuli were clouds of non-overlapping dots ($0.25°$ diameter, either light or dark grey with random proportion for every cluster, from a ratio of 20 : 80 to 80 : 20, Weber contrast 0.4), displayed at $12°$ eccentricity.

### (d) Reproduction task

On each trial, a reference dot cloud was first presented for 500 ms. After a 1 s pause, a second dot array appeared close to central fixation, which participants edited by trackpad to match as closely as possible the characteristics of the reference image: horizontal trackpad movements varied patch area, vertical movements varied density. Combined movements along the $+45°$ axis varied density and area together, hence also numerosity, while movements along the $-45°$ direction increased one while decreasing the other, keeping numerosity constant. The instructions (in Italian) were: 'Now you can see an image comprised of dots. Then a second one will appear; please adjust it to look as similar as possible to the image you first saw. To do this, edit the image by moving your finger left or right to make it larger or smaller, and up or down

to fill or empty it.' This response method was very natural even for the youngest children. All participants were allowed five practice trials (with no feedback) to familiarize them with the task and the set-up, which were excluded from the analysis.

Patterns were generated randomly at the beginning of each trial and dots added or removed at each trackpad movement. Reference arrays covered a circle with a radius of $5°$ and contained either 12 or 24 dots (randomly selected trial-by-trial). All children completed 2 sessions of 36 trials each.

### (e) 3AFC odd-one-out task

Three stimuli were presented simultaneously at the vertices of a virtual equilateral triangle. Two always contained 16 dots confined in a virtual circular zone of $3.6°$ radius. The other (the odd-one-out) differed in either area or density, and thus numerosity. Changes were selected trial-by-trial following the adaptive QUEST algorithm [18] and expressed in base-two logarithms of the ratio in area or density of the odd-one-out and reference stimuli. Both area and density could vary from the standard at most by 1.6 $\log_2$-units. When they varied together, the combined variation could not exceed 1.6 $\log_2$-units. Stimuli remained on the screen until the response (made by pressing an appropriate button on a custom-made response box). Instructions were: 'choose the one you think is different'. If children were hesitant, they were told: 'one of these images is different for some reason, choose the one you think is different'. The task began with eight training trials (excluded from analysis), structured so participants could familiarize themselves with the procedure without being cued for any of the variables. The training trials contained two examples of extremes along the density, area, numerosity or constant-numerosity axes, each $\pm 2$ $\log_2$-units from the reference stimulus. Feedback was given only about which was the correct target, not why it was correct. During practice, after children made their choice, an image of a flashing star appeared on screen in the place of the odd-one-out. Subsequent test trials were without feedback. All participants completed 100 to 120 test trials.

### (f) Numerosity discrimination

Two dot arrays were simultaneously presented left and right of a central fixation point, each array a virtual circle of $5°$ radius. One stimulus was fixed at 24 dots, the other varied in numerosity following the adaptive QUEST algorithm [18], and children chose the more numerous array. This task took place after the spontaneous paradigm tests (dot-array reproduction and odd-one-out) to avoid conditioning participants to pay particular attention to number. All children generally completed one session of 45 trials.

### (g) Mathematical abilities

All participants completed an age-standardized Italian paper-and-pencil battery for early mathematical abilities (TEDI-MATH test, 2015). The test was individually administered for either two sessions lasting 20 min or one 40 min session. To avoid inducing mathematical reasoning during psychophysics tasks, mathematical abilities were always tested at the end of the experimental session.

The battery covers a wide spectrum of mathematical competence: (i) Forward and backward verbal counting at intervals of one unit (ranges: 1–31, 0–9, 0–6, 3–10, 3–15, 5–9, 4–8 and 7–0; 15–0); (ii) forward counting at intervals of twos and tens; (iii) serial counting of images (animals) printed either randomly or orderly in virtual rows. At the end of each counting series children also reported how many images they counted (conservation task); (iv) construction of equivalent sets: children observed an image of seven tokens and had to recreate it by placing the right number of physical tokens on a plain sheet; (v) functional use of counting: children had to infer how many hats the experimenter has in his/her

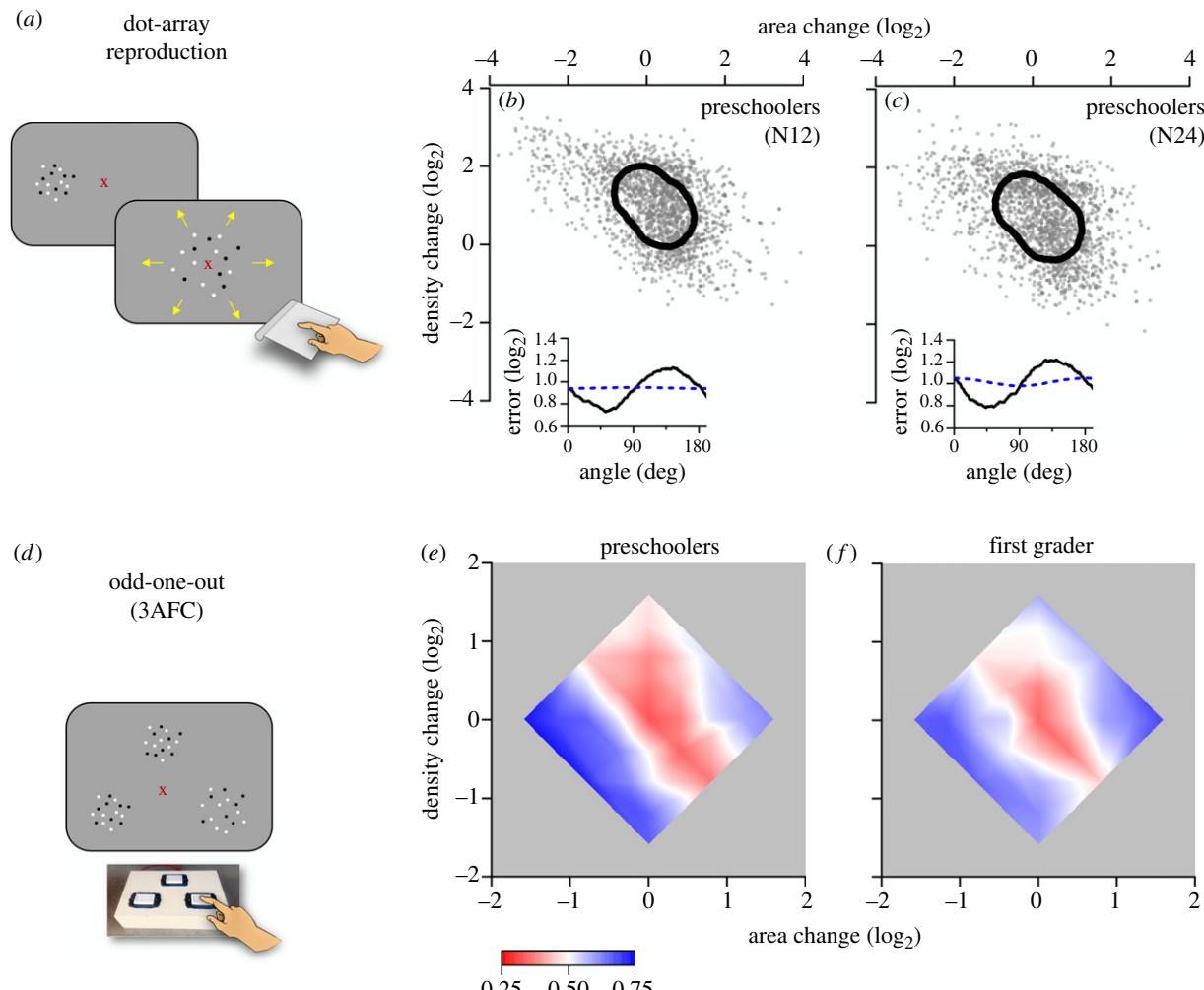

**Figure 1.** (*a*) Sample stimuli for dot-array reproduction task showing the two separate phases, presentation of sample and editing of matching stimulus. (*b,c*) Errors in the reproduction of numerosities 12 and 24 for the aggregate observer. Each dot corresponds to the response to a single trial; continuous lines indicate 84th percentile along each direction. The insets show the width of the error function along all directions in the area × density space. Black lines show data, blue dashed lines predicted behaviour of a mechanism that matches area and density independently, regardless of numerosity. (*d*) Illustration of 3AFC odd-one-out task. (*e,f*) Heat maps plotting per cent correct as a function of $\log_2$ of the normalized area and density of the target stimulus, separately for pre-schoolers and first graders. (Online version in colour.)

hand by counting the number of snowmen placed on the table; (vi) numeral comparison: children had to decide which of two Arabic numbers was numerically higher; (vii) oral decision: children had to decide if a verbally presented word was a number or not (non-targets were months, days or pseudo-words); (viii) seriation: children ordered tree patterns from the numerically smallest (1) to the higher (9); (ix) calculation: children had to mentally perform fast additions and subtractions of written digits (operations progressed in increasing difficulty from one digit to two digits); and (x) relative magnitude: children had to decide which of two numbers was numerically closer to a third.

Following the test manual, raw scores (number of correct responses) for each sub-test were calculated then converted into percentiles. For each participant, we then computed a summary mathematical-ability index by averaging the percentiles obtained in the separate sub-tests.

## (h) Data analysis

For the odd-one-out 3AFC task, we calculated the proportion of correct identification of the odd stimulus for all conditions, then linearly interpolated the data on a two-dimensional log–log space plotting normalized density against normalized area (figure 1). The data were fitted with a bi-dimensional Gaussian function, with three free parameters: orientation and lengths of the two axes. The orientation indicates the direction along

which subjects are most sensitive. The width of the minor axis is similar to the psychophysical just notable difference (JND) and measures the minimum variation along the axis of maximum sensitivity needed for the subject to reach 67% of correct responses (halfway between chance and perfect performance).

For the dot-array reproduction task, area and density of the reproduced patterns of each trial were plotted on a similar two-dimensional logarithmic space to that used for the odd-one-out task (figure 1), analysing responses separately for the two base numerosities 12 and 24. To obtain an aggregate observer, we lumped together the responses of the various individuals after first removing individual constant biases by subtracting their individual (two-dimensional) means, then adding the grand mean of the population to the distributions. For each dataset, we asked whether area and density were independent or correlated, by calculating the covariance matrix between the two dimensions. We then extracted the eigenvalues and the eigenvectors of the covariance matrices which correspond to the principal components of the data. The angle of maximal sensitivity corresponds to the angle of the second (shorter) principal axis, and thresholds are given by the standard deviation of the distribution along this direction.

For the 2AFC-numerosity-discrimination task, the proportion of trials where the test appeared more numerous than the probe was plotted against the logarithm of test-numerosity and fitted

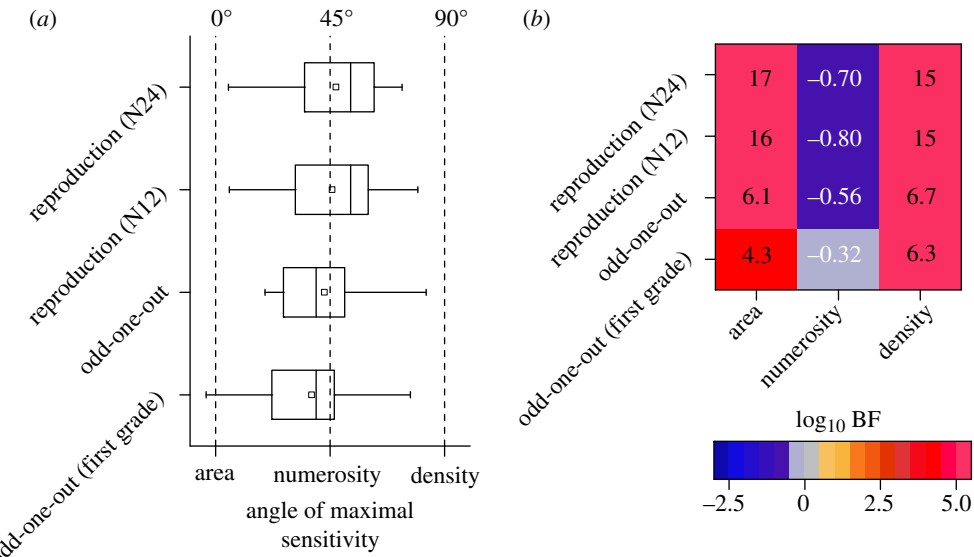

**Figure 2.** (*a*) Summary statistics for short-axis angle extracted from the error clouds of the reproduction task and heat maps of the odd-one-out task for individual participants. Boxes contain mean (square symbol), median (vertical line), 25th and 75th percentile. Whisker contains 5th and 95th percentile. (*b*) Log₁₀ Bayes factors (LBF) for Bayesian one sample *t*-tests (two-tailed) comparing data distributions against axes values predicted by area (0°), numerosity (45°) and density (90°) judgements. By convention, an LBF greater than 0.5, is considered substantial evidence in favour of the alternate hypothesis H₁, and LBF less than −0.5 substantial evidence in favour of the null hypothesis H₀. (Online version in colour.)

with cumulative Gaussian error functions. The 50% point of the error functions estimates the point of subjective equality (PSE), and the difference in numerosity between the 50 and 75% points gives the JND, which was used to measure thresholds as Weber fractions (JND/PSE).

Correlations between psychophysical test scores and mathematical ability were measured by non-parametric Kendall's tau ($\tau$). Statistical significance of correlations was reported by base-ten logarithms of Bayes factors (LBF). Bayes factors are an index of the likelihood that the experimental hypothesis is true, and in the simple case of no *a priori* knowledge equates to the ratio between these two likelihoods. LBFs close to zero indicate no substantial evidence gained from the current dataset; below −0.5 are considered substantial evidence in favour of the null hypothesis (H₀), above +0.5 are considered substantial evidence in favour of the alternative hypothesis (H₁) [19,20].

Comparison between regressions of explicit 2AFC and odd-one-out 3AFC thresholds with mathematical abilities was performed via bootstrap. On each of 100 000 iterations, we independently resampled participants (with replacement) and compared the correlation coefficients ($\tau$) of the two psychophysical scores with mathematics.

All data are available as part of the electronic supplementary material.

## 3. Results

### (a) Dot-cloud reproduction task in pre-schoolers

Fifty pre-schoolers performed the spontaneous reproduction task. After a brief presentation of a sample dot cloud, subjects were presented with a new dot cloud to edit with trackpad movements. They were not instructed what dimension they had to match but to create a pattern as similar as possible to that previously seen. Figure 1*b,c* reports reproduction errors for aggregate data. Each point represents the error in area (abscissa) and density (ordinate) for each individual trial for all participants. The contour line encircling the data indicates the dispersion along each direction at the 84th percentile. For

both test stimulus levels (N12 and N24), the response distribution tends to lie along the negative diagonal, the axis of constant numerosity, with the short axis near the numerosity axis: the angles were: 52.7° and 54.5° for N12 and N24. This suggests that children made smaller reproduction errors for variations in numerosity than for variations of area and density that did not lead to changes in numerosity.

The lower insets of the figure show the width of the response distribution along all directions in the area–density space. Consistent with inspection of the distributions, the direction where responses scatter least lies around 45° for both datasets. The blue lines show the prediction for a system blind to number that performed the reproduction task by matching area and density of the reference patch. Simulation parameters were chosen from a performance at the area and density directions (0° and 90°). The prediction is clearly a poor fit to the actual data, clear from inspection and by the explained variance ($R^2 = 0.01$ and $-0.04$ for N12 and N24).

The same analysis was performed on single-subject data, extracting for each participant the angle of maximal sensitivity as well as thresholds. Figure 2*a* reports group distributions of the angles of the shortest axis. For both numerosities (12 and 24 dots), distributions were centred near 45°, the prediction for spontaneous numerosity ($m = 45.8°$, s.d. = 22.9° and $m = 47.3°$, s.d. = 22.27° for N12 and N24, respectively). The angle of maximum sensitivity for the reproduction of the two numerosities correlated positively across participants, suggesting that the children maintained a consistent strategy ($\tau = 0.476$, LBF = 4.3). Figure 2*b* reports LBF for Bayesian one sample *t*-tests (two-tailed) comparing empirical data distributions against axes values predicted by reproduction of area (0°), numerosity (45°) and density (90°). The results showed that the hypothesis that the data come from a distribution aligned on the equal-numerosity axis was much more likely than the other options.

Average numerosity-discrimination thresholds were 0.45 (s.d. 0.17) and 0.52 (s.d. 0.22) log₂-units for 12 and 24

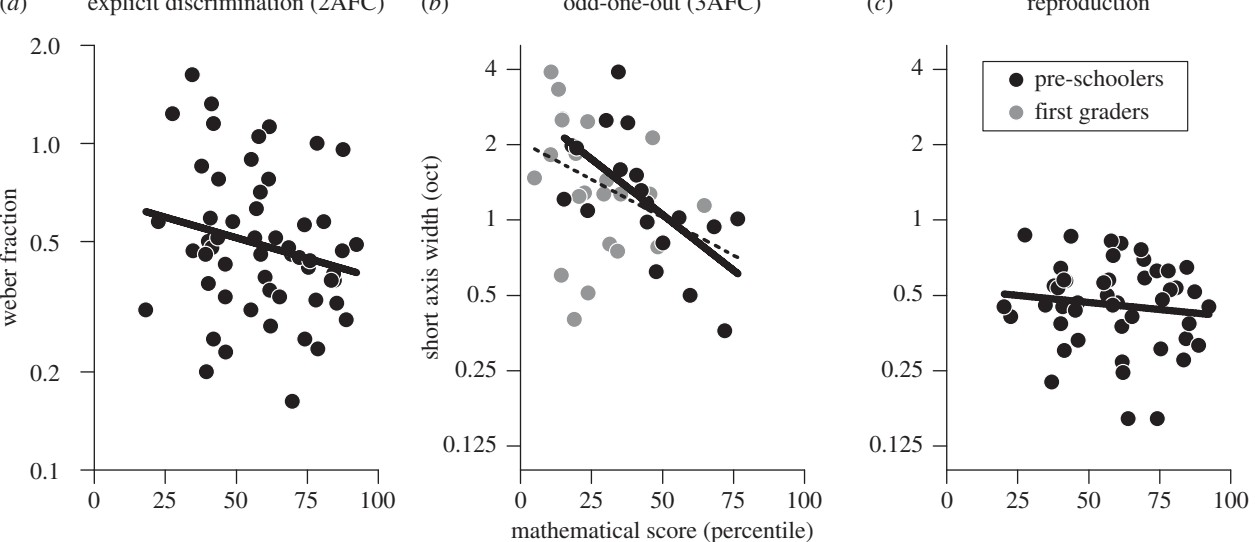

**Figure 3.** Correlation between mathematical ability scores and thresholds in 2AFC explicit numerosity discrimination (*a*), 3AFC odd-one-out task (*b*), and dot-array reproduction (averaging N12 and N24). Pre-schoolers are shown in black, first graders in grey. Thick lines are regressions through the pre-schooler group ($\tau = -0.171$, $-0.418$ and $-0.005$), dashed line in (*b*) are regressions through the whole group ($\tau = -0.29$). (Online version in colour.)

dot-stimuli, respectively. Average response times were 6.87 (s.d. 2.23) and 6.95 (s.d. 2.48) s.

## (b) Three-alternative-forced-choice odd-one-out task in pre-schoolers and first graders

A separate group of 20 pre-schoolers and 22 first graders were tested on another numerosity task, where they were presented with three simultaneous circular clusters of dots and asked to detect the 'odd' one. Two arrays had the same area and density (and therefore number, 16 dots), whereas a third stimulus differed in either area or density or both. Figure 1*e,f* plots proportion correct responses (pooled across subjects). As in the case of the reproduction task, the data are aligned to the equal-numerosity axis (negative diagonal), suggesting that the most sensitive dimension was again numerosity. The same analysis was also performed on single-subject data, from which we extracted the orientation of the short axis of the ellipses of each participant. Figure 2*a* shows that the short axes of most participants were scattered around the numerosity axis ($m = 42.7°$, s.d. $= 19.9°$; $m = 37.7°$, s.d. $= 26.0°$ for pre-schoolers and first-grade children, with no significant difference between groups $t_{40} = 0.69$, $p = 0.49$). LBF factors for Bayesian one sample *t*-tests (figure 2*b*) confirmed that data are very likely to fit a distribution centred on the numerosity prediction (45°), rather than predicted by area (0°) or density (90°).

Average thresholds were 1.15 log₂-units (s.d. 0.79) for pre-schoolers and 1.28 (s.d. 0.73) for first graders. Previous results obtained on adults under similar experimental conditions, provided thresholds around 0.65 log₂-units, again revealing a sharpening of the spontaneous number sense with age. The angle of maximal discrimination, however, remained almost stable at around 40° from 5 years to adulthood.

## (c) Correlations with early mathematical abilities

Previous studies have reported correlations between thresholds for explicit numerosity discriminations and mathematical abilities [1–9]. Here we tested whether this also holds for spontaneous tasks. All participants completed a mathematical-battery of age-standardized sub-tests. In order

to obtain a summary mathematical-ability index, percentiles were averaged across tests. Between subject mean percentile, across tests, was 48 (s.d. 21, min 11, max 92).

Figure 3*a* plots Weber fraction (measured with the traditional 2AFC technique) against mathematical score. The two are related, as has often been reported, but in this sample, the correlation was only marginally significant ($\tau = -0.171$ p $= 0.06$, LBF $= -0.03$ CI ($-0.35$, 0.01)). On the other hand, the width of the short axis measured in the 3AFC odd-one-out task correlated strongly with mathematics. Pooling over all pre-schoolers and first graders yielded a correlation of $\tau = -0.29$ ($p = 0.005$, LBF $= 0.956$, CI ($-0.51$: $-0.10$)), dashed line in figure 3*b*. Even considering only pre-schoolers, the correlation remained robust ($\tau = -0.418$, $p = 0.01$, LBF $= 0.74$, CI ($-0.66$: $-0.17$), continuous line in figure 3*b*). By bootstrap analysis, we compared in pre-schoolers the correlation strengths between mathematical ability and short axis in the odd-one-out task to that of the 2AFC explicit numerosity task. In 94% of iterations, the correlation coefficient for the odd-one-out task was stronger (i.e. more negative) than those of the traditional 2AFC, suggesting that the performance in the implicit 3AFC task tended to capture better the link with mathematics.

For the reproduction task neither the width of the short axes (figure 3*c*) nor response times were related to mathematical scores (thresholds: $\tau = 0.005$, LBF $= -0.72$; $\tau = -0.146$, LBF $= -0.28$, for test stimuli N12 and N24; response time: $\tau = 0.057$, LBF $= -0.65$; $\tau = -0.047$, LBF $= -0.67$). Short axis angles also did not relate to mathematics for both tasks (reproduction: $\tau = -0.013$, LBF $= -0.71$; $\tau = -0.06$, LBF $= -0.65$ for N12 and N24; 3-AFC task: $\tau = -0.08$, LBF $= -0.43$; $\tau = -0.177$, LBF $= -0.3$; $\tau = -0.156$, LBF $= -0.34$ for all participants, 5 year olds and first graders).

## 4. Discussion

This study examined whether numerosity (rather than area or density) spontaneously dominates the perception of young (5–6-year-old) children. We employed two spontaneous

psychophysical tasks previously used with older participants [16,17], in which subjects were asked either to identify the odd-one-out of a triplet or to reproduce the characteristics of a dot cloud. The results consistently showed that numerosity, rather than density or area, dominates perception, even without instructions or hints to base judgements on numerosity. Interestingly, the width of the short axes in the 3AFC odd-one-out task correlated well with mathematical scores, more strongly than did the traditional measurements of Weber fractions.

These results bear a strong similarity to those obtained for motion perception, where similar elongated response functions occur only in cases where dedicated mechanisms for motion detection have been described, such as vision [22], but not for audition [23]. Thus, our data strongly support the existence of a mechanism that compulsorily classifies very diverse stimuli on the bases of numerosity, even in young observers. Similar results have been obtained with pre-adolescent dyscalculics, suggesting that extended mathematical education is not needed to perceptually prioritize numerosity [17].

A recent study has examined how explicit numerosity judgements are perturbed by other non-numerical dimensions such as element size and surface area [24]. Several experimental groups (including primary school children, dyscalculics and mathematically-uneducated adults) all ignored the non-numerical dimensions, agreeing with this and previous [16,17,25] studies suggesting that numerosity is the spontaneous dimension. However, in Piazza et al.'s [24] study, the youngest participants, between 4 and 6 years old, did show a mild bias towards the non-numerical distractors, whereas our pre-schoolers showed adult-like behaviour at that age. However, the tasks of the two experiments were quite different, so a direct comparison is difficult. It is possible that the different behaviour in Piazza et al.'s study reflected a reduced capacity to inhibit irrelevant information.

Our data also reinforce the suggested link between the sensory resolution of the spontaneous number sense and pre-schooler mathematical scores. The 3AFC-spontaneous-task showed stronger correlations with mathematical scores than did the explicit numerosity comparison task. This suggests that the link between numerosity sensitivity and mathematics is based on numerosity resolution rather than on second-order mechanisms calculating number indirectly. This is consistent with our previous finding [8] that mathematics correlates with numerosity discrimination only at low densities, where numerosity mechanisms prevail, but not at high densities that may drive texture density mechanisms [26,27]. The link between numerosity and mathematics was more evident for the 3AFC odd-one-out rather than the reproduction task. This may have resulted from more individual variability in the reproduction task, possibly based on motor skills, which may have masked any existing correlation. Reproduction times and short-axis angles, for both perceptual tasks, were also not related to mathematics. However, both parameters were already at levels of typical developing adults and thus may have been saturated.

In conclusion, the results of this study suggest that as early as 5 years of age, before they have had any formal exposure to mathematical training, humans spontaneously perceive numerosity and that the link with mathematical skills is already present. These findings extend and support the idea that numerosity is a primary feature [16,28–30] and reinforce the hypothesis that it is an early building block for learning mathematical concepts and skills [12,31].

Ethics. The study was approved by the regional ethics committee at the *Azienda Ospedaliero-Universitaria Meyer* (protocol code: GR-2013-02358262). Parents signed the appropriate informed consent.

Data accessibility. The data that support the findings of this study are available as part of the electronic supplementary material.

Authors' contributions. G.M.C., G.A. and D.C.B. developed the study concepts. All authors contributed to the study design. Testing and data collection were performed by B.D.L., G.G. and M.M. G.M.C., G.A. and D.C.B. analysed and interpreted the data. G.M.C., G.A. and D.C.B. drafted the manuscript. All authors approved the final version of the manuscript for submission.

Competing interests. No authors have a potential conflict of interest.

Funding. This research was funded by the Italian Ministry of Health and by Tuscany Region under the project 'Ricerca Finalizzata', grant no. GR-2013–02358262 to G.A., from the European Research Council FP7-IDEAS-ERC (grant no. 338866—'Early Sensory Cortex Plasticity and Adaptability in Human Adults – ECSPLAIN), from European Research Council (ERC) under the European Union's Horizon 2020 research and innovation programme (grant agreement no. 801715—PUPILTRAITS and grant no. 832813—'Spatio-temporal mechanisms of generative perception'—GenPercept), from Italian Ministry of Education, University, and Research under the PRIN2017 programme (grant no. 2017XBJN4F—'EnvironMag' and grant no. 2017SBCPZY—'Temporal context in perception: serial dependence and rhythmic oscillations').

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
