## [Reviewer comments · Proceedings of the Royal Society B: Biological Sciences]

Review History

RSPB-2019-0770.R0 (Original submission)

Review form: Reviewer 1

Recommendation

Accept with minor revision (please list in comments)

Scientific importance: Is the manuscript an original and important contribution to its field?

Good

General interest: Is the paper of sufficient general interest?

Good

Quality of the paper: Is the overall quality of the paper suitable?

Excellent

Is the length of the paper justified?

Yes

Should the paper be seen by a specialist statistical reviewer?

No

Do you have any concerns about statistical analyses in this paper? If so, please specify them explicitly in your report.

No

It is a condition of publication that authors make their supporting data, code and materials available - either as supplementary material or hosted in an external repository. Please rate, if applicable, the supporting data on the following criteria.

Is it accessible?

Yes

Is it clear?

No

Is it adequate?

No

Do you have any ethical concerns with this paper?

No

Comments to the Author

This is a convincing paper using elegant psychophysical tasks to demonstrate that in young children (pre-schoolers and first graders) number is a variable that is spontaneously attended to (when it is contrasted with area and density).

I have asked myself why the authors decided to refer to their stimulus space (and results) in terms of octaves, instead of directly relating to base-two log-ratios. I suspect that the general reader of this kind of papers does not necessarily immediately grasp the correspondence between an octave and the base-two logarithm of the ratio across magnitudes. If the authors find it easier to use the notion of « octave » I suggest that they explicitly define what it refers to when they first introduce it.

I have one curiosity: for the 3AFC-odd-one-out task researchers have data in both pre-schoolers and first graders. Visual inspection of the accuracy distributions in the two groups (figure 1e-f) seem to suggest that in pre-schoolers there might a higher bias towards attending to density changes. However, this effect is not evident when the analysis is fitted on individual data (Figure 2a). Why is it the case?

Finally, I would appreciate that the authors briefly discuss/compare their results with those reported by a recent study by Piazza et al., 2018, that suggested that, in preschoolers, decisions based on number may suffer from high interference from other quantitative features of the sets. Because the results from the two studies could be interpreted as contrasting, it would be important to get some speculations on this point.

Minor. There is a wrong reference to the figures in the Results section.

Review form: Reviewer 2

Recommendation

Major revision is needed (please make suggestions in comments)

Scientific importance: Is the manuscript an original and important contribution to its field?
Excellent

General interest: Is the paper of sufficient general interest?
Excellent

Quality of the paper: Is the overall quality of the paper suitable?
Good

Is the length of the paper justified?
Yes

Should the paper be seen by a specialist statistical reviewer?
No

Do you have any concerns about statistical analyses in this paper? If so, please specify them explicitly in your report.
No

It is a condition of publication that authors make their supporting data, code and materials available - either as supplementary material or hosted in an external repository. Please rate, if applicable, the supporting data on the following criteria.

Is it accessible?
Yes

Is it clear?
Yes

Is it adequate?
Yes

Do you have any ethical concerns with this paper?
No

Comments to the Author

Manuscript ID RSPB-2019-0770

“Spontaneous perception of numerosity in pre-school children”

SUMMARY

Anobile et al test preschool and 1st grade children. Subsets of children were tested on 3 tests:

1. Reproduction task: the children alter a stimulus until it matches a previous seen sample
2. Odd-one-out task: they see 3 stimuli and have to indicate which is the odd one
3. Numerosity discrimination: the children have to choose the dot array with more dots

Test performance from test 2 strongly correlated with age appropriate math test scores. Authors conclude that children inherently focus on numerosity instead of area or density.

This is a very interesting and timely paper that nicely adds to the bulk of data published by the Burr laboratory showing a “number sense”. Having said that, there are several critical issues that need to be addressed prior to publication.

MAJOR CONCERNS

1. The manuscript is cumbersome to read; the authors change names of their tasks (e.g. “2AFC”

and “Numerosity discrimination”, “3AFC” and “odd-one-out”), the methods lack a lot of information, and the grammar/sentence structure should be improved (especially page 4, 3AFC). Please improve on these aspects.

2. The authors do show that numerosity perception seems to be more accurate than area or density perception. However, the authors also make the claim, that their study shows that preschool children use numerosity as primary feature, even without instructions or hints. Unfortunately, the authors did give hints: in the odd-one-out task, the authors do give feedback in training trials and thereby indirectly teaching the children to focus of numerosity. Please clarify how this can be reconciled with the idea of no-feedback.

3. The results of one of the other tasks (2AFC numerosity discrimination) are entirely missing.

MINOR CONCERNS

In the data repository are the following things missing:

- Numerosity discrimination data set is missing
- Math test performance data set is missing
- Description of the data; for example in “oddoneout”: what information is in “p”; what do the columns in “sceglidaqui” mean? Same holds for the other dataset.

The authors leave out crucial information which prohibits detailed understanding of their study:

- Page 3: “For the first analysis we aimed to measure the angle of maximal sensitivity with a 95% confidence interval to within 10°.” What do you mean with 10°? Same for “a SD 9°”, etc
- Page 8: “log Bayes factor”: how is this analysis working? It is not described in the methods.
- 3AFC task: two stimuli always comprised 16 dots and two stimuli always had the same area and density. Are these the same two stimuli?

Further analyses

- A large proportion of preschoolers participated in the 2AFC and reproduction task. Is there a correlation of performance in these tasks?
- The variance of age of the preschool children is quite large. Have you tested age as predictor for performance (or months spent in preschool)?
- Fig 1: You only show the variance in performance in respect to area and density. Since you claim that the children have higher accuracy for numerosity compared to area and density, it would be nice if you would show the same plots as in 1B,C with numerosity vs area, and numerosity vs density.
- A performance plot in % for each of the tasks would help to proper understand the results

FURHTER COMMENTS

- Methods, Participants: In the first sentence, they write that they tested 20 1st graders; later they write that 22 1st graders completed the odd-one-out task. Please correct.
- The other numbers of participants also don't add up: 58+20 preschooler != 72 preschooler; Therefore, the total number of participants must be wrong too.
- Page 4, reproduction task: inconsistent description of how dots can be added by the children. Is it by 45° axis movements or by mouse movement, or both? Please clarify.
- What is the QUEST algorithm?
- What do you mean with octaves? (e.g. page 4)
- Page 4: “...differed in either area or density, and thus numerosity” It is possible to keep area and density constant and only change numerosity...
- Page 5: “...This task took place after the spontaneous paradigm tests,...”. I cannot infer which of the two tests you mean. Please specify.
- In the results, Fig C-F is never referred to.
- Page 7, results: the first paragraph about the 2D Gaussian fit is not understandable.
- Reference 14: the link to the repository is not working
- Fig 1: What's the rationale behind showing the 84th percentile?
- Pictogram of the 2AFC task missing

Decision letter (RSPB-2019-0770.R0)

07-May-2019

Dear Professor Burr,

I am writing to inform you that your manuscript RSPB-2019-0770 entitled "Spontaneous perception of numerosity in pre-school children" has, in its current form, been rejected for publication in Proceedings B.

This action has been taken on the advice of referees, who have recommended that substantial revisions are necessary. With this in mind we would be happy to consider a resubmission, provided the comments of the referees are fully addressed. However please note that this is not a provisional acceptance.

Yours sincerely,
Loeske Kruuk
Editor

Proceedings B
<mailto:proceedingsb@royalsociety.org>

Associate Editor
Board Member: 1
Comments to Author:

We have now heard from two experts in the field, both of whom are largely positive. Both reviewers, however, raise some concerns. I am therefore recommending rejection. Nevertheless, I would encourage you to re-submit a new manuscript that deals with the comments raised.

Reviewer(s)' Comments to Author:

Referee: 1

Comments to the Author(s)

This is a convincing paper using elegant psychophysical tasks to demonstrate that in young children (pre-schoolers and first graders) number is a variable that is spontaneously attended to (when it is contrasted with area and density).

I have asked myself why the authors decided to refer to their stimulus space (and results) in terms of octaves, instead of directly relating to base-two log-ratios. I suspect that the general reader of this kind of papers does not necessarily immediately grasp the correspondence between an octave and the base-two logarithm of the ratio across magnitudes. If the authors find it easier to use the notion of « octave » I suggest that they explicitly define what it refers to when they first introduce it.

I have one curiosity: for the 3AFC-odd-one-out task researchers have data in both pre-schoolers and first graders. Visual inspection of the accuracy distributions in the two groups (figure 1e-f) seem to suggest that in pre-schoolers there might a higher bias towards attending to density changes. However, this effect is not evident when the analysis is fitted on individual data (Figure 2a). Why is it the case?

Finally, I would appreciate that the authors briefly discuss/compare their results with those reported by a recent study by Piazza et al., 2018, that suggested that, in preschoolers, decisions based on number may suffer from high interference from other quantitative features of the sets. Because the results from the two studies could be interpreted as contrasting, it would be important to get some speculations on this point.

Minor. There is a wrong reference to the figures in the Results section.

Referee: 2

Comments to the Author(s)

Manuscript ID RSPB-2019-0770

“Spontaneous perception of numerosity in pre-school children”

SUMMARY

Anobile et al test preschool and 1st grade children. Subsets of children were tested on 3 tests:

1. Reproduction task: the children alter a stimulus until it matches a previous seen sample
2. Odd-one-out task: they see 3 stimuli and have to indicate which is the odd one
3. Numerosity discrimination: the children have to choose the dot array with more dots

Test performance from test 2 strongly correlated with age appropriate math test scores. Authors conclude that children inherently focus on numerosity instead of area or density.

This is a very interesting and timely paper that nicely adds to the bulk of data published by the Burr laboratory showing a “number sense”. Having said that, there are several critical issues that need to be addressed prior to publication.

MAJOR CONCERNS

1. The manuscript is cumbersome to read; the authors change names of their tasks (e.g. “2AFC” and “Numerosity discrimination”, “3AFC” and “odd-one-out”), the methods lack a lot of information, and the grammar/sentence structure should be improved (especially page 4, 3AFC). Please improve on these aspects.
2. The authors do show that numerosity perception seems to be more accurate than area or density perception. However, the authors also make the claim, that their study shows that

preschool children use numerosity as primary feature, even without instructions or hints. Unfortunately, the authors did give hints: in the odd-one-out task, the authors do give feedback in training trials and thereby indirectly teaching the children to focus of numerosity. Please clarify how this can be reconciled with the idea of no-feedback.

3. The results of one of the other tasks (2AFC numerosity discrimination) are entirely missing.

MINOR CONCERNS

In the data repository are the following things missing:

- Numerosity discrimination data set is missing
- Math test performance data set is missing
- Description of the data; for example in "oddoneout": what information is in "p"; what do the columns in "sceglidaqui" mean? Same holds for the other dataset.

The authors leave out crucial information which prohibits detailed understanding of their study:

- Page 3: "For the first analysis we aimed to measure the angle of maximal sensitivity with a 95% confidence interval to within 10°." What do you mean with 10°? Same for "a SD 9°", etc
- Page 8: "log Bayes factor": how is this analysis working? It is not described in the methods.
- 3AFC task: two stimuli always comprised 16 dots and two stimuli always had the same area and density. Are these the same two stimuli?

Further analyses

- A large proportion of preschoolers participated in the 2AFC and reproduction task. Is there a correlation of performance in these tasks?
- The variance of age of the preschool children is quite large. Have you tested age as predictor for performance (or months spent in preschool)?
- Fig 1: You only show the variance in performance in respect to area and density. Since you claim that the children have higher accuracy for numerosity compared to area and density, it would be nice if you would show the same plots as in 1B,C with numerosity vs area, and numerosity vs density.
- A performance plot in % for each of the tasks would help to proper understand the results

FURHTER COMMENTS

- Methods, Participants: In the first sentence, they write that they tested 20 1st graders; later they write that 22 1st graders completed the odd-one-out task. Please correct.
- The other numbers of participants also don't add up: 58+20 preschooler != 72 preschooler; Therefore, the total number of participants must be wrong too.
- Page 4, reproduction task: inconsistent description of how dots can be added by the children. Is it by 45° axis movements or by mouse movement, or both? Please clarify.
- What is the QUEST algorithm?
- What do you mean with octaves? (e.g. page 4)
- Page 4: "...differed in either area or density, and thus numerosity" It is possible to keep area and density constant and only change numerosity...
- Page 5: "...This task took place after the spontaneous paradigm tests,...". I cannot infer which of the two tests you mean. Please specify.
- In the results, Fig C-F is never referred to.
- Page 7, results: the first paragraph about the 2D Gaussian fit is not understandable.
- Reference 14: the link to the repository is not working
- Fig 1: What's the rationale behind showing the 84th percentile?
- Pictogram of the 2AFC task missing

Author's Response to Decision Letter for (RSPB-2019-0770.R0)

See Appendix A.

RSPB-2019-1245.R0

Review form: Reviewer 2

Recommendation

Accept as is

Scientific importance: Is the manuscript an original and important contribution to its field?

Excellent

General interest: Is the paper of sufficient general interest?

Excellent

Quality of the paper: Is the overall quality of the paper suitable?

Excellent

Is the length of the paper justified?

Yes

Should the paper be seen by a specialist statistical reviewer?

No

Do you have any concerns about statistical analyses in this paper? If so, please specify them explicitly in your report.

No

It is a condition of publication that authors make their supporting data, code and materials available - either as supplementary material or hosted in an external repository. Please rate, if applicable, the supporting data on the following criteria.

Is it accessible?

Yes

Is it clear?

Yes

Is it adequate?

Yes

Do you have any ethical concerns with this paper?

No

Comments to the Author

The authors addressed all my concerns. I recommend the publication of this manuscript.

Decision letter (RSPB-2019-1245.R0)

17-Jun-2019

Dear Professor Burr

I am pleased to inform you that your Review manuscript RSPB-2019-1245 entitled "Spontaneous perception of numerosity in pre-school children" has been accepted for publication in Proceedings B.

The referee and Associate Editor do not recommend any further changes. Therefore, please proof-read your manuscript carefully and upload your final files for publication. Because the schedule for publication is very tight, it is a condition of publication that you submit the revised version of your manuscript within 7 days. If you do not think you will be able to meet this date please let me know immediately.

To upload your manuscript, log into <http://mc.manuscriptcentral.com/prsb> and enter your Author Centre, where you will find your manuscript title listed under "Manuscripts with Decisions." Under "Actions," click on "Create a Revision." Your manuscript number has been appended to denote a revision.

You will be unable to make your revisions on the originally submitted version of the manuscript. Instead, upload a new version through your Author Centre.

- 1) A text file of the manuscript (doc, txt, rtf or tex), including the references, tables (including captions) and figure captions. Please remove any tracked changes from the text before submission. PDF files are not an accepted format for the "Main Document".
- 2) A separate electronic file of each figure (tiff, EPS or print-quality PDF preferred). The format should be produced directly from original creation package, or original software format. Please note that PowerPoint files are not accepted.

- 3) Electronic supplementary material: this should be contained in a separate file from the main text and the file name should contain the author's name and journal name, e.g. `authorname_procb_ESM_figures.pdf`

All supplementary materials accompanying an accepted article will be treated as in their final form. They will be published alongside the paper on the journal website and posted on the online figshare repository. Files on figshare will be made available approximately one week before the accompanying article so that the supplementary material can be attributed a unique DOI. Please see: <https://royalsociety.org/journals/authors/author-guidelines/>

- 4) Data-Sharing and data citation

It is a condition of publication that data supporting your paper are made available. Data should be made available either in the electronic supplementary material or through an appropriate repository. Details of how to access data should be included in your paper. Please see <https://royalsociety.org/journals/ethics-policies/data-sharing-mining/> for more details.

<http://datadryad.org/submit?journalID=RSPB&manu=RSPB-2019-1245> which will take you to your unique entry in the Dryad repository.

Once again, thank you for submitting your manuscript to Proceedings B and I look forward to receiving your final version. If you have any questions at all, please do not hesitate to get in touch.

Sincerely,

Professor Loeske Kruuk
Editor
<mailto:proceedingsb@royalsociety.org>

Associate Editor
Board Member
Comments to Author:

I am pleased to recommend publication of your excellent manuscript. Congratulations.

Reviewer(s)' Comments to Author:

Referee: 2

Comments to the Author(s).

The authors addressed all my concerns. I recommend the publication of this manuscript.

Decision letter (RSPB-2019-1245.R1)

19-Jun-2019

Dear Professor Burr

I am pleased to inform you that your manuscript entitled "Spontaneous perception of numerosity in pre-school children" has been accepted for publication in Proceedings B.

Your article has been estimated as being 9 pages long. Our Production Office will be able to confirm the exact length at proof stage.

Open Access

Paper charges

Sincerely,

Appendix A

07-May-2019

Dear Professor Burr,

I am writing to inform you that your manuscript RSPB-2019-0770 entitled "Spontaneous perception of numerosity in pre-school children" has, in its current form, been rejected for publication in Proceedings B.

This action has been taken on the advice of referees, who have recommended that substantial revisions are necessary. With this in mind we would be happy to consider a resubmission, provided the comments of the referees are fully addressed. However please note that this is not a provisional acceptance.

Yours sincerely,
Loeske Kruuk
Editor

Proceedings B
mailto: proceedingsb@royalsociety.org

Associate Editor
Board Member: 1
Comments to Author:

We have now heard from two experts in the field, both of whom are largely positive. Both reviewers, however, raise some concerns. I am therefore recommending rejection. Nevertheless, I would encourage you to re-submit a new manuscript that deals with the comments raised.

Dear Editor,

We thank you for allowing to resubmit the manuscript. We thank the referees for the useful comments and we modified the ms accordingly; all changes are highlighted in blue.

Reviewer(s)' Comments to Author:

Referee: 1

Comments to the Author(s)

This is a convincing paper using elegant psychophysical tasks to demonstrate that in young children (pre-schoolers and first graders) number is a variable that is spontaneously attended to (when it is contrasted with area and density).

Thank you.

I have asked myself why the authors decided to refer to their stimulus space (and results) in terms of octaves, instead of directly relating to base-two log-ratios. I suspect that the general reader of this kind of papers does not necessarily immediately grasp the correspondence between an octave and the base-two logarithm of the ratio across magnitudes. If the authors find it easier to use the notion of « octave » I suggest that they explicitly define what it refers to when they first introduce it.

We thank the referee for this comment, and now refer to base-two logs rather than octaves.

I have one curiosity: for the 3AFC-odd-one-out task researchers have data in both pre-schoolers and first graders. Visual inspection of the accuracy distributions in the two groups (figure 1e-f) seem to suggest that in pre-schoolers there might a higher bias towards attending to density changes. However, this effect is not evident when the analysis is fitted on individual data (Figure 2a). Why is it the case?

Yes, with pre-schoolers there was a higher bias towards attending to density changes, with decision angles on average 42.7 for pre-schoolers and 37.7 for first grade children: however, this difference was not statistically different ($p=0.49$), now reported in the ms. This difference is hard to detect by visual inspection on Figure 2a, but it can be spotted by close inspection of the small change of the square symbol inside the box plot.

Finally, I would appreciate that the authors briefly discuss/compare their results with those reported by a recent study by Piazza et al., 2018, that suggested that, in preschoolers, decisions based on number may suffer from high interference from other quantitative features of the sets. Because the results from the two studies could be interpreted as contrasting, it would be important to get some speculations on this point.

Thank you for bringing this work to our attention. We read the paper carefully and believe it presents congruent rather than conflicting results. The two studies address two different

questions with different methodological approaches, but reached the same conclusion: numerosity dominates decisions over non-numerical information.

We now cite and discussed why there may be minor differences.

Minor. There is a wrong reference to the figures in the Results section.

Thank you, now fixed.

Referee: 2

Comments to the Author(s)

Manuscript ID RSPB-2019-0770

“Spontaneous perception of numerosity in pre-school children”

SUMMARY

Anobile et al test preschool and 1st grade children. Subsets of children were tested on 3 tests:

1. Reproduction task: the children alter a stimulus until it matches a previous seen sample
2. Odd-one-out task: they see 3 stimuli and have to indicate which is the odd one
3. Numerosity discrimination: the children have to choose the dot array with more dots

Test performance from test 2 strongly correlated with age appropriate math test scores.

Authors conclude that children inherently focus on numerosity instead of area or density.

This is a very interesting and timely paper that nicely adds to the bulk of data published by the Burr laboratory showing a “number sense”. Having said that, there are several critical issues that need to be addressed prior to publication.

MAJOR CONCERNS

1. The manuscript is cumbersome to read; the authors change names of their tasks (e.g. “2AFC” and “Numerosity discrimination”, “3AFC” and “odd-one-out”), the methods lack a lot of information, and the grammar/sentence structure should be improved (especially page 4, 3AFC). Please improve on these aspects.

Thank you. We have rewritten much of the manuscript with this comment in mind, hoping to make it easier to understand.

2. The authors do show that numerosity perception seems to be more accurate than area or density perception. However, the authors also make the claim, that their study shows that preschool children use numerosity as primary feature, even without instructions or hints. Unfortunately, the authors did give hints: in the odd-one-out task, the authors do give feedback in training trials and thereby indirectly teaching the children to focus on numerosity. Please clarify how this can be reconciled with the idea of no-feedback.

In the training trials (as during experimental trials) we never mentioned numerosity or other features; we just asked participants to indicate the odd stimulus. Importantly during training the feedback was given on the “oddity” and not numerosity.

We now state it more clearly in the ms.

3. The results of one of the other tasks (2AFC numerosity discrimination) are entirely missing.

The main aim of this work was to measure what variables drive participants' choices when inspecting dots arrays, so we omitted showing these results. However, we agree that is important to also report descriptive statistics of 2AFC thresholds, and have now added them to the ms. Please note that Figure 3A includes individual performance in this task.

MINOR CONCERNS

In the data repository are the following things missing:

- Numerosity discrimination data set is missing

Sorry about that, now added

- Math test performance data set is missing

Now added

- Description of the data; for example in “oddoneout”: what information is in “p”; what do the columns in “sceglidaqui” mean? Same holds for the other dataset.

We apologize to the referee for not providing a key to the variable names. We now include a separate word document which guides through all the datasets included.

The authors leave out crucial information which prohibits detailed understanding of their study:

- Page 3: “For the first analysis we aimed to measure the angle of maximal sensitivity with a 95% confidence interval to within 10° .” What do you mean with 10° ? Same for “a SD 9° ”, etc

We have now rephrased this section.

- Page 8: “log Bayes factor”: how is this analysis working? It is not described in the methods.

The Bayes factor (BF) is an alternative to the classical p-value. BF reflects the probability of the observed data under the null hypothesis compared to alternative hypothesis, allowing to quantify evidence in favour of the null hypothesis. We now added references to relevant work discussing how it is derived.

- 3AFC task: two stimuli always comprised 16 dots and two stimuli always had the same area and density. Are these the same two stimuli?

Yes, we now clarified this in the result section.

Further analyses

- A large proportion of preschoolers participated in the 2AFC and reproduction task. Is there a correlation of performance in these tasks?

Thank you for this suggestion. We ran the correlation and found that the tasks are well correlated. We now added this in the ms.

- The variance of age of the preschool children is quite large. Have you tested age as predictor for performance (or months spent in preschool)?

All correlations between age and perceptual performance (implicit but also explicit tasks) were far from significant (all $p > 0.2$). Please note that previous studies that have documented age development of numerosity sensory precision (WF) drew their analyses from larger developmental spans. The lack of correlation found here does not really demonstrate no correlation given our very narrow age range (pre-schoolers age range: 5–6.75), so we do not feel to take a strong position in the matter.

- Fig 1: You only show the variance in performance in respect to area and density. Since you claim that the children have higher accuracy for numerosity compared to area and density, it would be nice if you would show the same plots as in 1B,C with numerosity vs area, and numerosity vs density.

We thank the referee for this comment. We could plot number against the other dimensions, but this could be misleading, as the two quantities are not orthogonal. However, in order to help visualize the data more clearly, we have added a subplot that shows the width of response distribution along all possible dimensions. This plot shows two key features: first the direction with least spread is around 45 degrees, and that a system blind to numerosity that matched independently area and density would have produced a rather different data pattern.

- A performance plot in % for each of the tasks would help to properly understand the results.

In the dot-array reproduction task there was not a binary response and thus no % correct can be calculated. In the other two tasks, the stimulus intensity was chosen by a QUEST algorithm which enables optimal data collection. A side effect, however, is that different subjects are shown different stimuli, hence it is not possible to plot accuracy value at a pre-set stimulus intensity as not all the subjects may have that specific data point.

FURTHER COMMENTS

- Methods, Participants: In the first sentence, they write that they tested 20 1st graders; later they write that 22 1st graders completed the odd-one-out task. Please correct.

Now fixed.

- The other numbers of participants also don't add up: 58+20 preschooler != 72 preschooler; Therefore, the total number of participants must be wrong too.

Now fixed.

- Page 4, reproduction task: inconsistent description of how dots can be added by the children. Is it by 45° axis movements or by mouse movement, or both? Please clarify.

Now fixed.

- What is the QUEST algorithm?

The QUEST algorithm is an adaptive procedure for efficient threshold estimation. The algorithm is able to decide trial-by-trial, according to subject performance, the best stimulus intensity for the next trial, calculated as the maximum likelihood estimate of threshold. The algorithm is available as a free package within Psychtoolbox functions for MatLab environment and it is widely used in psychophysics experiment.

We now added the relevant reference.

Watson, A. B., & Pelli, D. G. (1983). QUEST: a Bayesian adaptive psychometric method. Percept Psychophys, 33(2), 113–120.

- What do you mean with octaves? (e.g. page 4)

The base-two log of the ratio between stimuli intensity (we now use base-two log in the ms).

- Page 4: "...differed in either area or density, and thus numerosity" It is possible to keep area and density constant and only change numerosity...

Please note that by our definition of density is number of items per unit area (N/area) thus keeping area constant, a numerosity change will inevitably change density.

- Page 5: "...This task took place after the spontaneous paradigm tests,...". I cannot infer which of the two tests you mean. Please specify.

Now added a specification there.

- In the results, Fig C-F is never referred to.

Thank you, now fixed.

- Page 7, results: the first paragraph about the 2D Gaussian fit is not understandable.

We now rephrase it.

- Reference 14: the link to the repository is not working

Now fixed.

- Fig 1: What's the rationale behind showing the 84th percentile?

The eight-fourth percentile corresponds to being 1 standard deviation away from the center of mass.

- Pictogram of the 2AFC task missing

As it is a standard task, we prefer not to add one more figure.